# Statistical Study of the Effectiveness of Surface Application of Graphene Oxide as a Coating for Concrete Protection

Andrea Antolín-Rodríguez [1], Daniel Merino-Maldonado [1], Álvaro Rodríguez-González [1], María Fernández-Raga [2], José Miguel González-Domínguez [3], Andrés Juan-Valdés [1] and Julia García-González [1,*]

1 Department of Engineering and Agricultural Sciences, School of Agricultural and Forest Engineering, University of Leon, Av. De Portugal 41, 24071 Leon, Spain
2 Department of Chemistry and Applied Physics, Industrial Engineering School, University of Leon, Vegazana Campus S/N, 24071 Leon, Spain
3 Instituto de Carboquímica (ICB-CSIC), Group of Carbon Nanostructures and Nanotechnology (G-CNN), C/Miguel Luesma Castán 4, 50018 Zaragoza, Spain
* Correspondence: julia.garcia@unileon.es

**Abstract:** Improving the protection of concrete by applying graphene oxide (GO) as a surface treatment has become the objective of the present study. This study focuses on performing a statistical analysis to study different levels of GO application as an exterior coating, thus observing the effectiveness of the coating and the optimization of the treatment material for concrete protection. Four tests were performed to define concrete durability, such as pressurized water penetration, capillary absorption, freeze-thaw resistance and carbonation resistance. The results showed an increase in concrete durability with any level of GO application on the surface, considering that the optimum amount of application for water impermeability and freeze-thaw resistance is 26.2 μg/cm², since it was possible to reduce pressurized water penetration by 45%, capillary water absorption by 57% and freeze-thaw detachment by 25%. However, the optimum application rate for carbonation resistance is 52.4 μg/cm², reducing carbonation by almost 60%. In conclusion, if the concrete is going to be exposed to less aggressive environments, the application of a mild surface coating of GO is sufficient for its protection, and if the concrete is going to be exposed to more aggressive environments, it is necessary to increase the amount of GO. The performance of GO as a coating significantly increased the degree of protection of the concrete, increasing its service life and proving to be a promising treatment for concrete surface protection.

**Keywords:** graphene oxide; statistical analysis; coating; durability; concrete

## 1. Introduction

Throughout their useful life, concrete structures are exposed to different environmental conditions that cause their degradation and deterioration. Generally, this deterioration is caused by various agents (carbonation, chlorides, etc.) that lead to a shortening of the service life of the concrete elements [1]. For this reason, protection against degradation has gained significant attention, since this problem has caused great damage in the construction industry, being also the origin of costly repairs or even replacements with the consequent economic and environmental cost [2].

The most commonly employed strategy in recent years has been the use of high-performance concretes, which are distinguished by their low permeability and high strength. However, new more economical approaches are being sought that focus on providing additional protection to the materials [3,4], such as the use of corrosion inhibitors or surface treatment [5,6]. Nowadays, surface treatment is one of the most widely accepted approaches due to its effectiveness in preventing the entry of aggressive substances from the environment and protecting not only against corrosion inside the structures, improving

their mechanical behavior, but also protecting the whole set that integrates the cementitious matrix, thus extending its usefulness as a consequence of a greater durability of the treated elements [7].

Recently, one of the most promising methods is the use of nanomaterials [8], thanks to their extraordinary potential to penetrate through cracks and pores in concrete [9]. So far, the literature has focused little on the study of surface treatments with nanomaterials [10–14].

In this study, a graphene-based material has been used as a possible protective treatment for concrete with the objective of studying whether it can be a potential candidate for its preservation, taking into account its cost with other coatings and the range of protection it offers.

Graphene oxide (GO) is an excellent nanomaterial, as it has successively demonstrated important functions and broad prospects in the field of coating fabrication [15]. So far, GO has been employed as a coating modifier material to improve the barrier and strength of coatings, such as GO and ETEO-modified epoxy resin coatings, GO-modified silane emulsions, GO/ITBS coatings or GO-modified epoxy coatings [16–20], due to its unique properties and highly specific surface area [20,21]. However, GO as a unique coating material has been scarcely reported in the literature [22,23], being this a good starting point for its study. In this regard, one of the most relevant properties of 2D GO assemblies should be highlighted; it has been shown that metal ions ($Al^{3+}$, $Fe^{3+}$, $Ca^{3+}$, $Mg^{2+}$, etc.) are excellent crosslinking agents for GO membranes, making them stronger and more resistant to washing [24]. These alkaline earth cations are some of the main components of cement used in the manufacture of concrete, so GO could be an excellent protective coating for concrete due to a possible chemical bonding to its surface by virtue of the aforementioned cations.

However, in addition to studying GO spraying as a surface coating of concrete, it is also necessary, for practical reasons, to establish the optimal amount while keeping intact its degree of protection, which is the objective of this article. There is not any report, to the best of our knowledge, that establishes the optimum protective coating for concrete; therefore, a statistical study is conducted to determine the surface effectiveness of GO as a protective coating. Consequently, water transmission, resistance to freeze/thaw cycles and resistance to carbonation were studied; all of them with defining parameters of concrete durability. Concrete specimens without any coating and concrete specimens with various amounts of GO surface coatings were evaluated. The results of this research provide further information on the suitability of a GO coating as a protective surface treatment for concrete, as well as to optimize the amount of GO required for an efficient concrete protection, always with a view to achieving a "functional coating," that is, a coating that is economically viable and increases the durability of the concrete, taking into account the optimization of resources.

## 2. Materials and Methods

### 2.1. Materials

#### 2.1.1. Hardened Concrete

The cement used was blast furnace slag CEM IIIA 42.5 N/SR, which complies with the requirements of EN 197-1 [25]. The aggregates used were of a siliceous nature, using 4/12.5 mm gravel as coarse aggregate and 0/4 mm sand as fine aggregate. These aggregates are considered suitable for concrete production in accordance with EN 12620:2003+A1 [26] and the Eurocode [27].

The dosage applied for the manufacture of the concrete complies with the mechanical and durability requirements of the European standards (EN) and is presented in Table 1.

**Table 1.** Concrete mixture components.

| Components (Quantity/m$^3$) | Conventional Concrete |
| --- | --- |
| Gravel (kg) | 1030.7 |
| Sand (kg) | 650.5 |
| Cement (kg) | 390.0 |
| Water (L) | 198.0 |

Concrete batching was established according to the De La Peña method, based on the desired characteristic strength of the final concrete [28].

### 2.1.2. Graphene Oxide
Synthesis of GO

The GO used for the surface coating of the hardened concrete samples in the present study was obtained by exfoliation of the graphitic oxide in water and subsequent application of mild ultrasound. Graphitic oxide was produced by an oxidation method known as the Hummers method, with certain particular modifications. The complete experimental process for obtaining graphene oxide is reported in [23], as well as the complete characterization of graphitic oxide and GO thereof.

Figure 1 shows a transmission electron microscopy (TEM) image of freshly exfoliated graphene oxide, showing single-layer thick GO sheets of approximately (2–4) μm lateral size. The two-dimensional and thin nature of this material allows its physical and flexible adaptability, thus confirming that GO is a convenient material at micro-nanoscale to act as a protective coating.

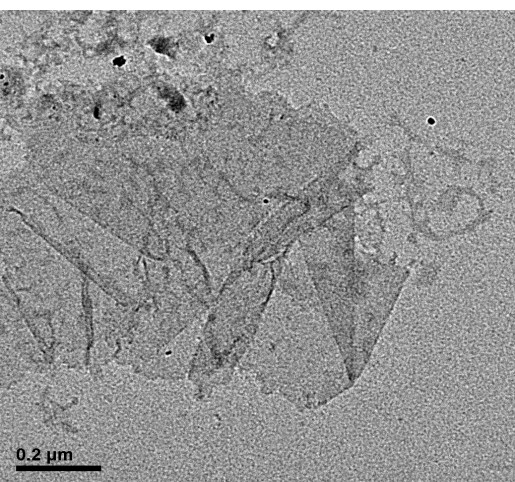

**Figure 1.** TEM image of a GO flake. Recorded with a JEOL-200FXII (JEOL, Tokyo, Japan) microscope working at 200 kV and with 0.28 nm and with 0.28 nm point-to-point resolution (Electron Microscopy Sciences, ref CF-400CU).

### 2.2. Treatment

The treatment consists of the application of a dispersion of GO in aqueous suspension (0.5 mg/mL), without any other additives or adjuvants, over the surface of the hardened concrete. The treatment is applied by means of an airbrush with horizontal movement and slow speed, controlling the amount of treatment deposited on the surface (26.2 μg/cm$^2$ per each coating passage). The application time between passages is set at 1 h and after the complete treatment, the concrete specimens are left to dry at a temperature of $20 \pm 2$ °C and a relative humidity of $45 \pm 15$ % for two days.

The concrete samples were cured for 90 days at a temperature of $20 \pm 2$ °C and relative humidity >95% prior to surface application of the GO coating.

### 2.3. Experimental Design

The experimental design included four trials with six treatment levels and three replicates of each treatment (72 samples in total). The treatments analyzed were the following: one treatment with one GO coating (1-GO coating), one treatment with two GO coatings (2-GO coatings), one treatment with three GO coatings (3-GO coatings), one treatment with four GO coatings (4-GO coatings), one treatment with five GO coatings (5-GO coatings) and one control treatment, that is, uncoated concrete specimens, without any treatment (CC). Table 2 lists each treatment level and the amount of GO contained in each.

**Table 2.** Treatment levels and surface content of each level.

| Treatment Levels (Coating) | GO Content ($\mu g \cdot cm^2$) |
|---|---|
| CC | 0.0 |
| 1-GO coating | 26.2 |
| 2-GO coatings | 52.4 |
| 3-GO coatings | 78.6 |
| 4-GO coatings | 104.8 |
| 5-GO coatings | 131.1 |

### 2.4. Methods

The research method involved the development of four tests focused on the durability assessment of concrete. Subsequently, a statistical analysis was performed to each of the tests, with an aim at evaluating the minimum significant differences between the treatment levels, thus determining which is the minimum GO surface concentration necessary for the full concrete protection to occur.

#### 2.4.1. Depth of Water Penetration under Pressure

Standard cylindrical concrete specimens (150 mm Ø × 300 mm height) were tested to determine the penetration depth of pressurized water according to UNE-EN 12390-8 [29]. The test was conducted over a period of 72 h in a water penetration equipment, which allows a hydrostatic pressure of 5 bar (0.5 MPa) on the upper base of the specimen. After this period of pressurized water application, the specimens are broken by splitting tensile strength according to the protocol established in the EN 12390-6 [30] standard, which allows observing the water penetration front, which goes from the base towards the interior of the cylindrical specimen. For the calculation of the area enclosed by the penetration front, the ImageJ software program is used, which allows the calculation of areas. To determine the average depth of penetration ($P_m$), Equation (1) contained in the UNE-EN 12390-8 [29] standard is used, written as follows:

$$P_m = \frac{Apf}{d} \tag{1}$$

where the area of the penetration front $A_{pf}$ ($mm^2$) was divided by the specimen diameter $d$ (mm), thus obtaining the average penetration depth $P_m$ (mm).

#### 2.4.2. Water Absorption by Capillarity

Cubic specimens (100 mm × 100 mm × 100 mm) were tested for capillary water absorption according to UNE 83982 [31]. The test consisted of placing the concrete specimens, after conditioning according to the UNE 83966 [32] standard, in a plastic container with a leveling grid, on which the specimens were placed in contact with a layer of deionized water about 5 mm high. The capillary absorption coefficient (K) was determined by means of the following Equation (2), contained in the UNE 83982 [31] standard:

$$K = \frac{\delta a \cdot \varepsilon_e}{10 \cdot \sqrt{m}} \tag{2}$$

where K is the capillary absorption coefficient (kg/m$^2 \cdot$ min$^{0.5}$), $\varepsilon_e$ is the effective porosity of concrete (cm$^3$/cm$^3$), $\delta_a$ is the density of water (the value of 1 g/cm$^3$ is considered) and m is the resistance to water penetration by capillary absorption (min/cm$^2$).

### 2.4.3. Resistance to Freezing/Thawing with De-Icing Salts

Truncated conical specimens (110 mm Ø $\times$ 75 mm Ø $\times$ 85 mm height) were tested with a layer of (5 $\pm$ 2) mm of a 3% NaCl solution in potable water as de-icing salt. The assembly is subjected to 28 freeze-thaw cycles according to EN 1339-Anex D [33], which each cycle lasting 24 h alternating temperature decreases from (20 $\pm$ 5) °C to ($-20 \pm 5$) °C for 17 h, and temperature increases from ($-20 \pm 5$) °C to (20 $\pm$ 5) °C for 7 h. The freeze-thaw resistance of the hardened concrete was determined by recording the weight of material released per unit area (kg/m$^2$).

### 2.4.4. Resistance to Carbonation at Atmospheric Levels of $CO_2$

Hardened concrete cubic specimens (100 mm $\times$ 100 mm $\times$ 100 mm) are exposed to a natural environment for a period of 6 months and protected from direct rain as described in EN 12390-10 [34]. After the exposure period, the specimens were divided by tensile strength (EN 12390-6 [30]) and a phenolphthalein solution was sprayed. Once the solution was applied and following the provisions of standard EN 12390-10 [34], the carbonation depth was determined perpendicular to the surface of the concrete specimen, using a caliper. Three central points were measured, located at 0.25, 0.5 and 0.75 of the edge length. From the calculated carbonation depth, the carbonation resistance of the concrete was determined by calculating the carbonation speed according to the following Equation (3):

$$V_{CO_2} = \frac{x}{\sqrt{t}} \tag{3}$$

where $V_{CO_2}$ is the carbonation speed (mm/year$^{0.5}$), $x$ is the carbonation depth (mm) and $t$ is the exposure time (years).

### 2.4.5. Statistical Analysis

The data were subjected to an ANOVA one-way to comparison of means, considering the durability parameters (pressurized water penetration depth, capillary water absorption, freeze/thaw resistance and carbonation resistance) as variable factors and GO treatment level as a fixed factor. Differences ($p \leq 0.05$) between durability parameters and between GO treatment levels were examined by comparison of means using the LSD (Least Significant Difference) post-hoc test.

All analyses were performed with the SPSS Software (Statistical Package for the Social Sciences) version 21 (IBM, Statistics, SPSS Inc., Chicago, IL, USA).

## 3. Results and Discussion

The four parameters analyzed indicated that the increase in the protection of concrete was conditioned by the level of treatment applied to its surface, i.e., the surface concentration of deposited GO, showing a direct correlation between the protective efficacy and the amount of coating. However, when studying the increase in concrete protection, it is necessary to consider the optimization of the treatment material, so the optimal treatment level for concrete protection was determined from the statistical analysis performed for each durability parameter. The final decision about the quantity to add should take into consideration also the cost that the treatment has in comparison with the improvement on the durability parameters. So, it was selected per each use the minimum product that will represent a significative difference in properties.

### 3.1. Depth of Penetration of Water under Pressure

The greatest depth of water penetration under pressure (20.9 mm) occurs in the uncoated control samples (CC). The application of any GO coating on the surface of the

concrete specimens implies a decrease in the water penetration depth, there being a direct relationship between the decrease in the water penetration depth and the amount of GO applied on the surface of the concrete. The results of the statistical analysis showed that the 1-GO treatment, which is the first level of treatment, presented significant differences with the CC samples, which are the samples without any coating, since the penetration depth of the 1-GO samples (11.4 mm) was significantly lower (F = 42.951; df = 5.12; $p \leq 0.001$) than that of the control samples (20.8 mm), implying a decrease of 45% (Figure 2).

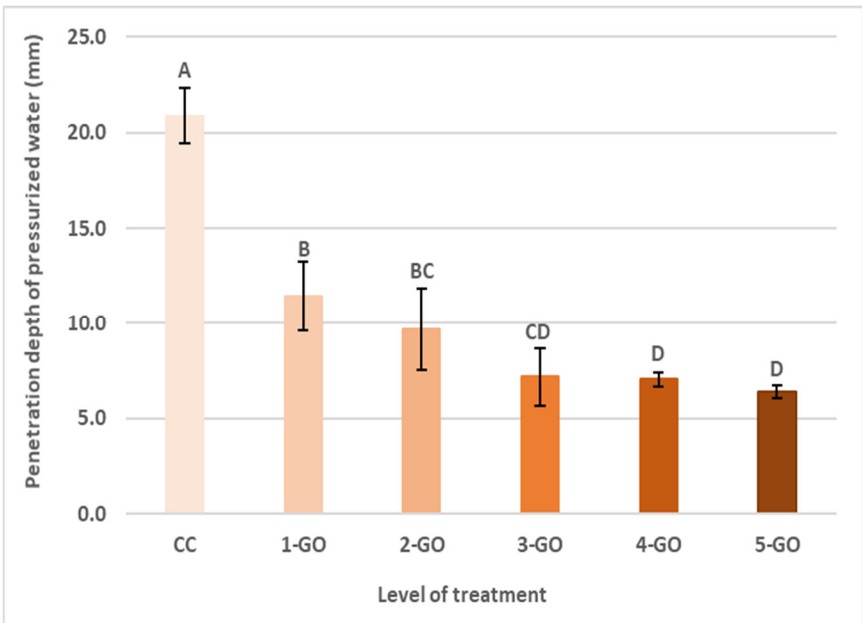

**Figure 2.** Depth of water penetration under pressure at different treatment levels. Capital letters indicate minimum significant differences between treatment levels. Vertical bars represent the mean and the standard error (SE). Statistical differences are indicated by different letters (LSD test, $p < 0.05$).

The 1-GO treatment showed no significant differences with the subsequent level of treatment (2-GO), with the application of the 3-GO treatment being necessary for minimal significant differences to exist. The samples with three GO coating passages (3-GO) showed a water penetration depth of 7.2 mm, which implies a decrease of 37% with respect to the 1-GO samples, but an increase in the amount of GO applied on the surface of the concrete samples of 52.4 $\mu$g/cm$^2$. Therefore, it is established that the improvement in protection achieved with one GO coating passage is sufficient, considering it as the optimal level for the protection of the concrete surface against the penetration of pressurized water.

### 3.2. Capillary Absorption

The highest capillary water absorption coefficient (0.035 kg/m$^2$min$^{0.5}$) is observed in the control samples (CC), without any type of coating. The application of any level of GO coating on the surface of the concrete samples, implies a decrease in the capillary absorption coefficient, there being a direct relationship between the decrease in the "K" coefficient and the amount of GO deposited on the surface of the concrete. Regarding the results obtained in the statistical analysis of the capillary absorption coefficient, it is established that the first level of treatment, compared to those samples without any type of coating, presented significant differences, since the capillary absorption coefficient was significantly lower for the 1-GO treatment (F = 29.090; df = 5.12; $p \leq 0.001$) than that obtained in the control samples, being 0.015 kg/m$^2$min$^{0.5}$, which represents a decrease of 57% (Figure 3).

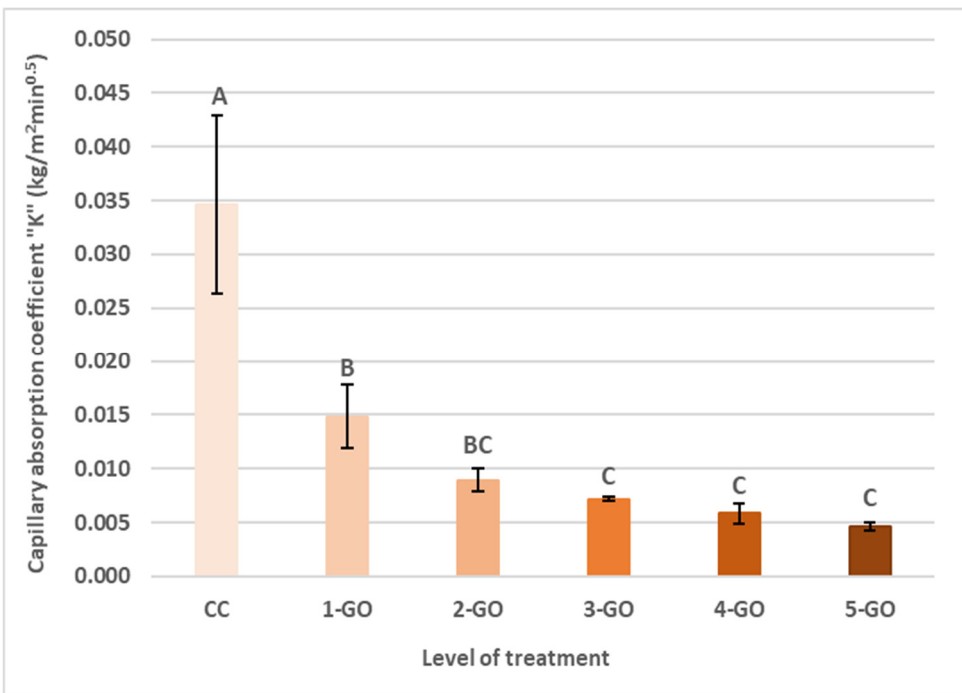

**Figure 3.** Coefficient of capillary water absorption at different treatment levels. Capital letters indicate minimum significant differences between treatment levels. Vertical bars represent the mean and the standard error (SE). Statistical differences are indicated by different letters (LSD test, $p < 0.05$).

The 1-GO treatment showed no significant differences with the 2-GO treatment, which is the subsequent level of treatment, with the application of the 3-GO treatment being necessary for minimal significant differences to exist. The 3-GO samples showed a k coefficient of 0.007 kg/m$^2$min$^{0.5}$, which implies a decrease of 52% with respect to the 1-GO samples, but a neat increase in the amount of GO applied of 52.4 μg/cm$^2$. Therefore, it is established that the improvement in concrete protection achieved with 1-GO coating is sufficient, considering it the optimal level for the protection of concrete against capillary water absorption.

*3.3. Freeze/Thaw Resistance*

The greatest mass loss due to freeze/thaw cycles occurs in uncoated specimens (CC), with a mass loss of 6.8 kg/m$^2$ in this type of sample. The application of any GO coating treatment implies an increase in the resistance of the concrete to freeze/thaw cycles by decreasing the amount of material detached from the surface; there being a direct relationship between the decrease in such amount of material detached and the amount of GO deposited on the surface of the concrete. Looking at Figure 4, the statistical analysis performed to evaluate freeze/thaw resistance showed that the first treatment level with which the CC samples showed significant differences was the 1-GO treatment, since the mass loss is significantly lower (F = 9.752; df = 5.12; $p$ = 0.001) than that obtained in the CC samples, 5.1 kg/m$^2$, implying a 25% decrease.

The 1-GO treatment showed no significant differences with the subsequent two treatment levels (2-GO and 3-GO), with the application of the 4-GO treatment being necessary for minimal significant differences to exist. The specimens with four GO coating passages (4-GO) showed a mass loss of 4 kg/m$^2$, which implies a decrease of 22% with respect to the 1-GO specimens, but an increase in the amount of GO applied on the concrete surface of 78.6 μg/cm$^2$. Therefore, it is established that the improvement in protection achieved with one GO coating passage is sufficient, being considered as the optimal level for the protection of the concrete surface against freezing/thawing.

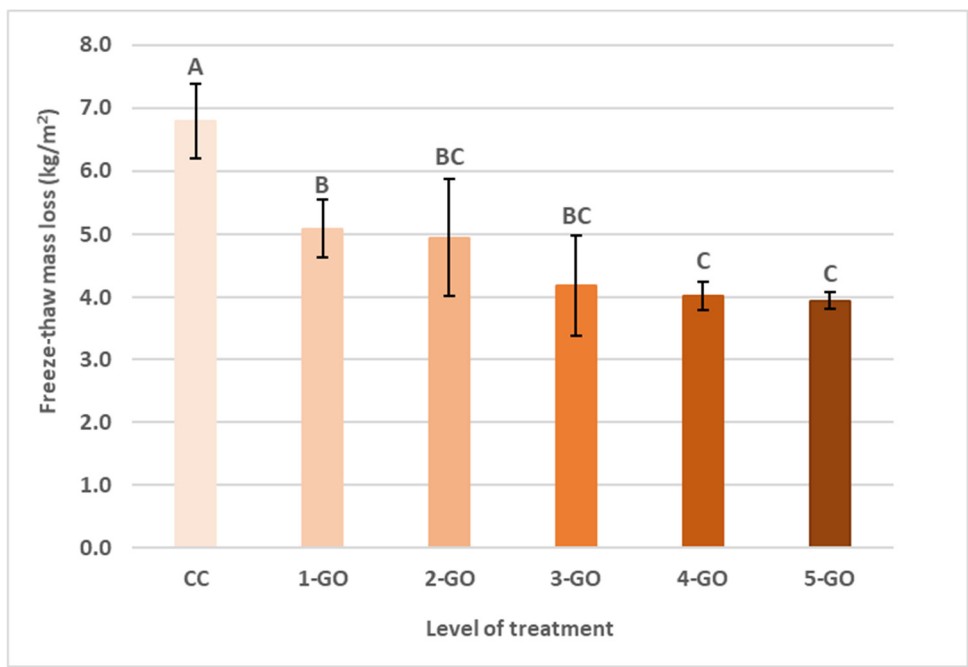

**Figure 4.** Mass loss after freeze-thawing cycles at different treatment levels. Capital letters indicate minimum significant differences between treatment levels. Vertical bars represent mean and standard error (SE). Statistical differences are indicated by different letters (LSD test, $p < 0.05$).

*3.4. Resistance to Carbonation*

The highest carbonation speed is found in the samples without any type of coating (CC), with 7.4 mm/year$^{0.5}$. The application of any GO coating treatment implies an increase in the resistance of the concrete to carbonation, as the carbonation speed decreases; there is a direct relationship between the decrease in the carbonation speed and the amount of GO deposited on the surface of the concrete. As displayed in Figure 5, the results of the statistical analysis indicated that the 1-GO treatment is the first treatment with which the CC samples presented significant differences, since the carbonation speed of this treatment was significantly lower (F = 15.096; df = 5.12; $p \leq 0.001$) than that produced in the CC samples, 5.0 mm/ year$^{0.5}$, which represents a decrease of 32%.

The 1-GO treatment showed significant differences with the subsequent treatment level (2-GO), presenting a carbonation speed of 3.0 mm/ year$^{0.5}$, which implies a 40% decrease compared to the 1-GO specimens. The 2-GO treatment did not show significant differences with the rest of the higher treatment levels (3-GO, 4-GO and 5-GO). Therefore, it is established that the improvement in protection achieved with a GO coating is not enough, being necessary to reach two GO coating passages, considering this as the optimal level for the protection of the concrete surface against the carbonation process.

From all the results obtained in the four-durability test, it is observed that for water impermeability of the concrete surface and for freeze/thaw resistance the optimal amount of coating is 26.2 μg·cm$^2$, while for carbonation resistance the optimal amount of coating is 52.4 μg·cm$^2$. This difference may be justified by the type of exposure to which the concrete is subjected to, in each test, since different kinds of concrete exposure may produce slight changes with respect to durability requirements. According to Eurocode [27], in the water penetration and capillary absorption test, the concrete is in exposure X0, which corresponds to a kind of exposure without risk of corrosion. In the freeze/thaw test, the concrete in in exposure XF2, which corresponds to an exposure class for freeze/thaw attack in which the concrete is moderately saturated with melting salts. Finally, in the carbonation resistance test, the concrete is in an XC3 exposure, which corresponds to a carbonation-induced corrosion exposure kind in which the concrete is subjected to natural exposure with medium-high humidity but protected from rain.

Exposures X0 and XF2 involve less aggressive environments and therefore the optimal amount of protection is 26.2 $\mu g \cdot cm^2$; however, exposure XC3 involves a much more aggressive environment and the application of 26.2 $\mu g \cdot cm^2$ is not sufficient and it is necessary to extend the optimum amount up to 52.4 $\mu g \cdot cm^2$.

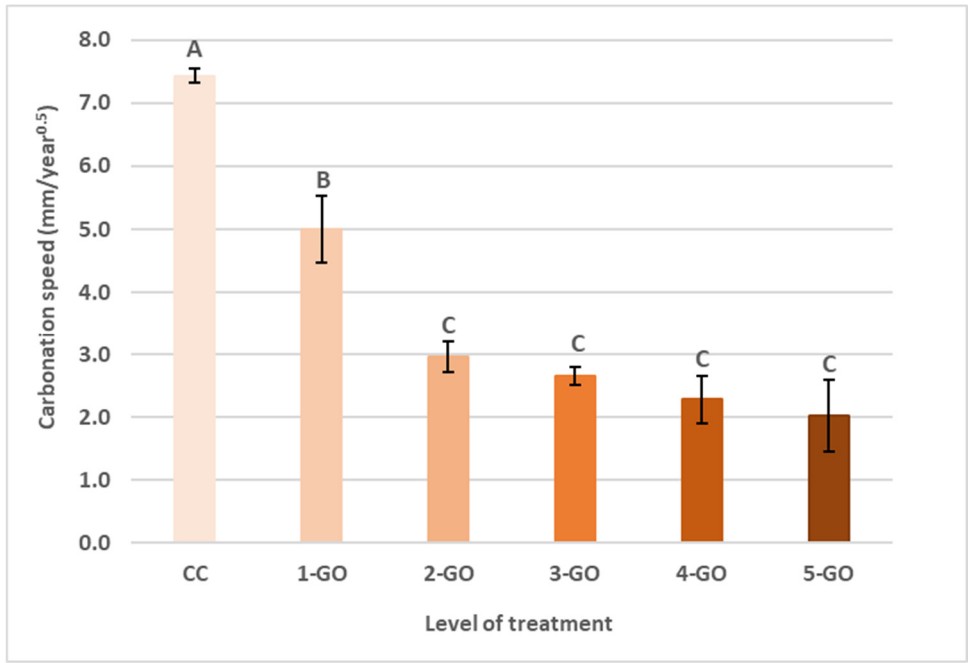

**Figure 5.** Carbonation speed at different treatment levels after 6 months exposure. Capital letters indicate minimum significant differences between treatment levels. Vertical bars represent mean and standard error (SE). Statistical differences are indicated by different letters (LSD test, $p < 0.05$).

## 4. Conclusions

The application of a GO coating improved the protection of concrete surfaces by increasing its durability. The results of the durability tests showed that any application of GO on the surface of the concrete resulted in an increase in its protection and with a direct correlation between the protection efficacy and the deposited amount of GO. However, it is necessary to determine an optimal coating extent, balancing between the effective protection and the optimization of the material used treatment. Therefore, the results of the statistical analysis showed that the optimal GO surface concentration for the water impermeability and freeze-thaw resistance tests was 26.2 $\mu g \cdot cm^2$, while for the carbonation resistance it was 52.4 $\mu g \cdot cm^2$.

In conclusion, if the concrete is going to be exposed to less aggressive environments, the application of a mild surface coating of GO (26.2 $\mu g \cdot cm^2$) is sufficient for its protection, if the concrete is going to be exposed to more aggressive environments, such as carbonation, it is necessary to increase the amount of GO (52.4 $\mu g \cdot cm^2$), to maintain an adequate protection of the concrete.

In addition, GO dispersion is commercially widely available and relatively affordable, as it is the cheapest graphene derivative available, being Spain and Europe where it is produced the most. Currently, a GO dispersion with a concentration of 0.05% wt is commercially available at around 70 €/liter. Therefore, taking into account the economic cost of this product, the optimization of the product that has been studied and the protection it provides to the concrete, GO can be accepted as an excellent surface treatment, which can be practical for large surfaces or specific applications due to the small amount used during its application.

**Author Contributions:** Conceptualization, A.J.-V., J.G.-G. and A.A.-R.; methodology, A.A.-R., Á.R.-G. and D.M.-M.; software, A.A.-R.; formal analysis, A.A.-R., J.M.G.-D., M.F.-R. and J.G.-G.; investigation, A.A.-R., D.M.-M. and Á.R.-G.; resources, J.M.G.-D. and M.F.-R.; data curation, A.A.-R., Á.R.-G. and D.M.-M.; writing—original draft preparation, A.A.-R., A.J.-V., Á.R.-G. and J.G.-G.; writing—review and editing, A.A.-R., A.J.-V., J.G.-G., J.M.G.-D., M.F.-R. and Á.R.-G.; supervision, J.G.-G. and A.J.-V. All authors have read and agreed to the published version of the manuscript.

**Funding:** A.A.R. greatfully acknowledge to Aid from the Junta de Castilla y Leon to finance the pre-doctoral hiring of research personnel, co-financed by the European Social Fund, resolved in the ORDER EDU/875/2021, of July 13. D.M.M. greatfully acknowledge to Aid from the Junta de Castilla y Leon to finance the pre-doctoral hiring of research personnel, co-financed by the European Social Fund, resolved in the ORDEN EDU/601/2020, of July 3. M.F.R and J.M.G.-D. greatfully acknowledge Spanish Ministry of Science and Innovation (MICINN) and the Spanish Research Agency (AEI) for the financial support provided by the NANOSHIELD project (PID2020-120439RA-I00).

**Institutional Review Board Statement:** Not applicable.

**Informed Consent Statement:** Not applicable.

**Data Availability Statement:** Not applicable.

**Conflicts of Interest:** The authors declare no conflict of interest.

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
