# Peer review of "Statistical Study of the Effectiveness of Surface Application of Graphene Oxide as a Coating for Concrete Protection"

_coatings, doi:10.3390/coatings13010213_

Round 1

Reviewer 1 Report

line 91, 104, 129, 205, 252 etc. - Error! ...

What mean symbols A, B, BC, CD and D in Figs 2 and 3

Are there some relationships between depth of water penetration under pressure at different treatment levels (Fig. 2) and between coefficient of capillary water absorption at different treatment levels (Fig. 3) and other relationships in Figs?

Author Response

Reviewer #1:

Correct “Error! Reference source not found” found in lines 91, 104, 129, 205, 232, 252 and 274.

We have successfully made this change.

What mean symbols A, B, BC, CD and D in Figs 2 and 3.

The letters show the minimum significant differences that exist between treatment levels, indicating the mean value from highest to lowest. When there are different letters between the treatment levels, it means that there are minimum significant differences between the treatment levels; on the contrary, when the share the same letter, it means that there are no minimum significant differences between them. The interactions that exist between all the treatment levels have been analysed.

Figures 2, 3, 4 and 5, which show the results of the various tests carried out, already show in their description that the statistical differences are indicated by different letters.

Are there some relationships between depth of water penetration under pressure at different treatment levels (Fig 2.) and between coefficient of capillary water absorption at different treatment levels (Fig 3.) and other relationship in Figs?

No, the parameters are independent, each providing a different measure of durability, they serve together to study the behaviour of concrete under different conditions.

Thank you very much for your comments and suggestions on our manuscript.

On behalf of all the co-authors, we thank you in advance for your time and kind attention.

Reviewer 2 Report

This study is presenting novel results on graphene oxide coatings on concrete surface that serves as protection from atmospheric influences and increases durability. There is no plagiarism in the text, the text is concise and clear and the English is good.

Only several remarks are seen:

-          Please, define the abbreviation GO at the first mention.

-          Delete the “Error!” messages within the text and add missing data.

-          Statistics performed is not quite clear, neither the letters presented in figures. Why were the linear models observed? It is advisable to show correlations of the obtained data in a separate section.

-          How expensive is GO, is it affordable and practical for the large concrete surfaces, or can it be used in some specific applications. Please, add this to the manuscript.

Author Response

Reviewer #2:

This study is presenting novel results on graphene oxide coatings on concrete surface that serves as protection from atmospheric influences and increases durability. There is no plagiarism in the text, the text is concise and clear and the English is good.

Only several remarks are seen:

-          Please, define the abbreviation GO at the first mention.

We have successfully made this comment.

-          Delete the “Error!” messages within the text and add missing data.

We have successfully made this modification.

-          Statistics performed is not quite clear, neither the letters presented in figures. Why were the linear models observed? It is advisable to show correlations of the obtained data in a separate section.

There was a typo when describing the statistical method used, in this case the date were subjected to a one-way ANOVA. This statistical method does not show correlations between the data. This modification has been recorded by means of the change control in the “Statistical Analysis” section.

On the other hand, the letters represent the statistical differences that exist between treatment levels, as determined in the figure caption of each of the results. The letters show the minimum significant differences that exist between treatment levels, indicating the mean value from highest to lowest. When there are different letters between the treatment levels, it means that there are minimum significant differences between these treatment levels; on the contrary, when they share the seam letter, it means that there are no minimum significant differences between them.

-          How expensive is GO, is it affordable and practical for the large concrete surfaces, or can it be used in some specific applications. Please, add this to the manuscript.

This information has been added to the manuscript, in the conclusions section.

Thank you very much for your comments and suggestions on our manuscript.

On behalf of all the co-authors, we thank you in advance for your time and kind attention.

Reviewer 3 Report

presented in Er-91 ror! Reference source not found.. >>> reference citation errors should be fixed

Hardened Concrete >>> samples should be well defined

Concrete mixture components.  >>>code/standard should be defined for design mix criteria

The abstract should briefly state the purpose of the research, the principal results, and major conclusions. An abstract is often presented separately from the article, so it must be able to stand alone.

The necessity and innovation of the article should be presented to the introduction.

Author Response

Reviewer #3:

Presented in Er-91 ror! Reference source not found… >>> reference citation errors should be fixed.

We have successfully made this modification.

Hardened Concrete >>> simples should be well defined

The simples are well defined in the explanation of each of the tests that have been carried out, it was considered to define here the simples since each test presents a different type of sample. Which is determined in the standard corresponding to each of the tests.

Thus, in the water penetration depth test under perssure, the concrete simples used were cylindrical with dimensions of 150 mm in diameter and 300 mm in height. In the capillary test, the concrete simples used were cubic, 100 mm high x 100 mm wide x 100 mm long. Fort he freeze/thaw resistance test, the simples used were truncated cone-shaped, 110 mm top diameter x 75 mm bottom diameter and 85 mm sample height. Finally, the carbonation resistance test showed that the specimens used were cubic in shape, 100 mm high x 100 mm wide x 100 mm long.

Concrete mixture components >>> code/standard should be defined for design mix criteria

There is no European reference code/standard for defining design mix criterio. It is true, however, that information collected in the literature is used to establish the mix design.  In this case, the Peña criteria were used. This information has been added to the manuscript.

The abstract should briefly state the purpose of the research, the principal results, and major conclusions. An abstract is often presented separately from the article, so ir must be able to stand alone.

The abstract already briefly indicates the objective of the research when it states that "the study focuses on improving the protection of concrete by applying graphene oxide as a surface treatment, focusing the study on a statistical analysis to study various levels of application".

The main results were also indicated, as it was stated within the abstract that there was an increase in the durability of concrete with any application of GO on the surface, also determining the optimum amount depending on the test and what improved durability.

As for the main conclusions, it should be added in the abstract that if the concrete is going to be exposed to less aggressive environments, the application of a surface coating of GO is sufficient for its protection, while if it is going to be exposed to more aggressive environments, such as carbonation, it is necessary to increase the amount of GO to double. This information is now included in the manuscript.

The necessity and innovation of the article should be presented to the introduction.

                The need for the article is reflected in the introduction, since it establishes the search for new, more economical approaches focused on providing additional protection to materials, especially based on the study of new surface treatments, which is precisely what this article focuses on due to the existing need.

On the other hand, the innovation of the article is also reflected since it includes the promising use of nanomaterials as surface treatment, as well as the almost inexistent bibliography that exists in this regard.

Thank you very much for your comments and suggestions on our manuscript.

On behalf of all the co-authors, we thank you in advance for your time and kind attention.

Reviewer 4 Report

The topic of the article is interesting and should be presented in more detail. The various Sections, in fact, are excessively synthetic and this sometimes makes it difficult to reconstruct the entire process of acquiring the experimental results. The Authors should provide more details, even adding some more figures and/or some formulas.

Authors should also check their cross-references carefully (table 1, figure 1, etc.).

Line 106

“as well as its physical and flexible adaptability”

How would the transmission electron microscope (TEM) image reflect physical and flexible adaptability? Are the Authors alluding to the fact that the film seems to form several folds without breaking? The Authors should be more specific in their statements.

Lines 145-147

“the area of the penetration front Apf (mm2) was calculated and divided by the specimen diameter d (mm), thus obtaining the average penetration depth Pm (mm).”

This is not clear. The area of the penetration front is the product of the specimen height and the penetration depth, which is not constant along the specimen height. Therefore, to calculate the average penetration depth (in mm) it is necessary to divide the area of the penetration front by the specimen height (and possibly by 2, if the penetration front is measured from both sides), not by the specimen diameter. The average penetration depth can then be divided by the diameter of the specimen to obtain the average percentage of penetration (in percent). Please clarify this point.

What average penetration depth values were actually used in Figure 2?

Lines 170-171

“the depth of the carbonation front was determined by spraying a phenolphthalein solution.”

More precisely, the depth of the carbonation front was determined after spraying a phenolphthalein solution, since spraying a phenolphthalein solution is not a measurement method. How did you then actually measure the depth of the carbonation front? Did you measure an area in this case too? If so, by what linear dimension was the area divided?

Line 320

“while of the concrete is going to be exposed to more aggressive environments”

Do the Authors mean “IF the concrete …”?

Author Response

Reviewer #4:

The topic of the article is interesting and should be presented in more detail. The various Sections, in fact, are excessively synthetic and this sometimes makes it difficult to reconstruct the entire process of acquiring the experimental results. The Authors should provide more details, even adding some more figures and/or some formulas.

Authors should also check their cross-references carefully (table 1, figure 1, etc.).

We have successfully made this modification.

Line 106: “as well as its physical and flexible adaptability”

How would the transmission electron microscope (TEM) image reflect physical and flexible adaptability? Are the Authors alluding to the fact that the film seems to form several folds without breaking? The Authors should be more specific in their statements.

 Thanks to TEM image, which single-layer-thick Go sheets of a few (2-4) µm lateral size are observed. The very thin sheets allow for flexible and physical adaptability.

Lines 145-147: “the area of the penetration front Apf (mm2) was calculated and divided by the specimen diameter d (mm), thus obtaining the average penetration depth Pm (mm).”

This is not clear. The area of the penetration front is the product of the specimen height and the penetration depth, which is not constant along the specimen height. Therefore, to calculate the average penetration depth (in mm) it is necessary to divide the area of the penetration front by the specimen height (and possibly by 2, if the penetration front is measured from both sides), not by the specimen diameter. The average penetration depth can then be divided by the diameter of the specimen to obtain the average percentage of penetration (in percent). Please clarify this point.

What average penetration depth values were actually used in Figure 2?

 This information has been explained again in the manuscript in more detail.

The average penetration depth was measured according to UNE-EN 12390-8.

Lines 170-171: “the depth of the carbonation front was determined by spraying a phenolphthalein solution.”

More precisely, the depth of the carbonation front was determined after spraying a phenolphthalein solution, since spraying a phenolphthalein solution is not a measurement method. How did you then actually measure the depth of the carbonation front? Did you measure an area in this case too? If so, by what linear dimension was the area divided?

This information has been explained again in the manuscript in more detail.

The carbonation depth was measured according to EN 12390-10.

 “The depth of carbonation should be determined by turning according to the method given below. The position of the carbonation front should be measured at three points on each face. To locate these points, the length of the points, the length of the edge should be divided into four equal distances. The three central points, i.e., the points at 0.25, 0.5 and 0.75 of the rim length, should be used as measuring points.

With the aid of a ruler or caliper and a magnifying glass, the depth of perpendicular carbonation (dk) should be determined.”

Line 320: “while of the concrete is going to be exposed to more aggressive environments” Do the Authors mean “IF the concrete …”?

Yes, that´s what the authors meant. It has already been modified in the manuscript.

Thank you very much for your comments and suggestions on our manuscript.

On behalf of all the co-authors, we thank you in advance for your time and kind attention.

Round 2

Reviewer 4 Report

Response to the Comment on line 106 (of the original document)

Authors are encouraged to include the response in the revised paper.

Equation 1 (of the new document)

Based on this equation, penetration probably proceeds from one of the two bases into the inside of the cylindrical specimen. As this is not clear from the text and the reader is not required to know the reference norm, Authors should specify it in Section 2.4.1.

Author Response

Response to the Comment on line 106 (of the original document). Authors are encouraged to include the response in the revised paper.

The response has been included in the revised document, and can be checked through the change control.

Equation 1 (of the new document). Based on this equation, penetration probably proceeds from one of the two bases into the inside of the cylindrical specimen. As this is not clear from the text and the reader is not required to know the reference norm, Authors should specify it in Section 2.4.1.

The authors have included in Section 2.4.1. that the penetration front goes from the base towards the inside of the cylindrical specimen, as well as that the tested face is the upper base of the specimen.

Thank you very much for your comments and suggestions on our manuscript.

On behalf of all the co-authors, we thank you in advance for your time and kind attention.

Round 3

Reviewer 4 Report

The Authors addressed all of the comments of the reviewer. The manuscript is now suitable for publication in Coatings.